# Effect of Exogenous Calcium on the Heat Tolerance in *Rosa hybrida* 'Carolla'

Han Wang [1], Yuxiao Shen [1], Kaixuan Wang [1], Songlin He [1,2], Wan-Soon Kim [3], Wenqian Shang [1], Zheng Wang [1,*] and Liyun Shi [1,*]

1    College of Landscape Architecture and Art, Henan Agricultural University, Zhengzhou Wenhua Road 95, Zhengzhou 450002, China
2    College of Horticulture Landscape Architecture, Henan Institute of Science and Technology, Xinxiang 453003, China
3    Department of Environmental Horticulture, University of Seoul, Seoul 130-743, Korea
*    Correspondence: zwang17835268270@163.com (Z.W.); sisyrin@henau.edu.cn (L.S.);
     Tel.: +86-136-4384-6891 (Z.W.); +86-135-2661-5636 (L.S.)

**Abstract:** This study was designed to investigate the effects of exogenous calcium on the tolerance of *Rosa hybrida* 'Carolla' to high-temperature and the physiological mechanisms underlying this response. Leaves of 'Carolla' grown under stress were treated by spraying four different concentrations of calcium chloride ($CaCl_2$; 50, 100, 150, or 200 μM). The photosynthetic responses, antioxidant enzyme activities, and osmotic substance contents were measured under high-temperature stress at the flowering stage. Temperature-stressed 'Carolla' with $CaCl_2$ treatment showed significantly increased chlorophyll (Chl) content, net photosynthetic rate (*An*), transpiration rate (E), stomatal conductance (*gs*), water use efficiency (WUE), superoxide dismutase (SOD), peroxidase (POD), and ascorbate peroxidase (APX) activities together with proline (Pro), soluble sugar (SS), and soluble protein (SP) concentrations, while malonaldehyde (MDA) content and relative electrical conductivity (REC) were significantly reduced. The damages caused by high-temperature stress were alleviated by applying $CaCl_2$. Among the $CaCl_2$ treatments, 100 μM $CaCl_2$ best minimized the damage caused by high-temperature to 'Carolla'. This study showed that exogenous calcium could improve the tolerance of *Rosa hybrida* 'Carolla' to high-temperature stress by regulating photosynthesis, the antioxidant system, and osmotic substances.

**Keywords:** $CaCl_2$; cut rose; high-temperature tolerance; photosynthesis; physiological mechanism

## 1. Introduction

High temperatures affect the growth and development of plants, which is mainly reflected in morpho-anatomical, physiological, and biochemical changes [1]. High temperatures in summer can lead to growth restriction, a shorter flowering period, and lower ornamental quality [2]. The high-temperature inhibits the photosynthesis [3], potentially causing the accumulation of reactive oxygen species (ROS) such as hydrogen peroxide ($H_2O_2$) and malonaldehyde (MDA) [4], which disrupts membrane stability [5]. For roses, temperature ranging from 15 to 28 °C is appropriate for the growth and flower production [6]. Heat stress is considered when plants are exposed to temperatures of 5–15 °C above suitable growth conditions for about 15 min to a few hours [7]. However, temperatures in most regions of China are usually higher than 30 °C in summer, often exceeding 35 °C or even 40 °C in July and August [3].

With its pure color, straight flower stems, long flowering period, and strong disease resistance, *Rosa hybrida* 'Carolla' is one of the largest selling varieties of cut flowers on the market [8]. Cut roses exhibit recurrent flowering year-round under suitable temperature [9]; however, unsuitable growing conditions such as high temperatures can affect its growth status and ornamental quality. In roses, high-temperature can lead to abnormal plant

development, such as a bent neck (bent peduncle phenomenon, BPP) [10], decreased leaf area [11], leaf yellowing, withering and drop [12], flower deformity and disruption of flower bud differentiation, and other physiological diseases [13], which seriously affect the ornamental quality and commodity value of roses. Previous studies in 22 cultivars of commercial garden roses showed that the stomatal conductance ($gs$) and transpiration rate (E) of 'Gentle Hermione' is the lowest, while the chlorophyll (Chl) content of 'William Shakespeare 2000' is the lowest at a high-temperature of 38 °C [14]. The high-temperature increased the MDA content and relative electrolyte leakage (REL) of garden roses 'Marie Curie' and 'Lapjau' [15]. MDA and REC are both indicators of lipid peroxidation [16], indicating that high-temperature damages plant cellular membranes [5]. Studies have shown that heat stress has negative effects on the antioxidant system of three cultivars of wild roses [17], including the enzymatic system that protects plants from ROS damage. Early responses to heat stress include increases in the activities of superoxide dismutase (SOD) and peroxidase (POD) [3], but later inhibition of the antioxidant enzyme system leads to decreased enzyme activity [18]. Another study of the *Rosa chinensis* 'Old Blush' heterogeneous grafting line showed that high-temperature during the first nine hours causes a decrease in the levels of osmotic substances such as proline, soluble sugars, and soluble protein [12], which decreases heat tolerance. Proline, soluble sugars, and soluble protein are important osmoprotectants, and their abundance is positively correlated with stress resistance [16].

The exogenous application of plant growth regulators alleviates the adverse effects of high temperatures by improving the heat tolerance of plants [4]; calcium ($Ca^{2+}$) has also been shown to ameliorate the adverse effects of heat stress on plants [16]. Indicators of the heat stress response include photosynthetic responses, antioxidant enzyme activities, cell membrane integrity [19], and the content of osmotic substances. It has been shown that exogenous $Ca^{2+}$ increased the heat tolerance of *Arabidopsis* [20] and *Rhododendron* [21], which might be associated with altering carbohydrate metabolism [22], delays the loss of chlorophyll [23], higher antioxidant enzyme activities [24], and decreased electrolyte leakage [25]. However, excessive $Ca^{2+}$ concentration might be cytotoxic [24], and the optimal concentration of $Ca^{2+}$ needs further experiments. This study used these indexes to explore whether exogenous calcium can minimize heat stress damage and improve heat tolerance of the *Rosa hybrida* 'Carolla' by spraying 0, 50, 100, 150, or 200 µM $CaCl_2$ on leaves of 'Carolla' grown under high-temperature. Therefore, investigating the physiological mechanism of *Rosa hybrida* in response to heat stress after exogenous $Ca^{2+}$ has important theoretical and practical significance for applying the optimal $Ca^{2+}$ concentration to improve the heat tolerance of *Rosa hybrida* and achieve the annual production and application of *Rosa hybrida*.

## 2. Materials and Methods

### 2.1. Plant Material and Treatments

The experiment was conducted in controlled environment chambers (Thermoline TPG-6000-TH) at Henan Agricultural University. Two-year-old *Rosa hybrida* 'Carolla' plants were pruned uniformly before treatment, three well-grown buds were kept on each plant and sampled when the branches were about 50 cm long and each branch had 7–9 leaves. The day/night temperatures during the experiment were 35/20 °C, and relative humidity (RH) was maintained at approximately 70–80%. Light irradiance was up to 850 µmol m$^{-2}$ s$^{-1}$ under a 12/12 h light/dark cycle. The nutrient solution was dissolved in water and irrigated daily. The nutrient solution was composed of $Ca(NO_3)_2 \cdot 4H_2O$, $KNO_3$, EDTA-Fe, $Mg(NO_3)_2 \cdot 6H_2O$, $(NH_4)_2PO_4$, $MnSO_4 \cdot 5H_2O$, $ZnSO_4 \cdot 7H_2O$, $H_3BO_3$, $CuSO_4 \cdot 5H_2O$, and $(NH_4)_6Mo_7O_{24} \cdot 4H_2O$, which provided 1.841, 2.323, 64.5, 204.8, 575, 12.05, 8.63, 9.27, 1.25, and 0.88 mg L$^{-1}$, and $H_2SO_4$ provided 281 µL L$^{-1}$, respectively (EC 1.0 dS m$^{-1}$, pH 6.0 ± 0.2). Pest and disease control were applied as required.

One week after the bud break, the first calcium chloride ($CaCl_2$) sprays were carried out on the plant leaves. The foliar applications were conducted at 10-day intervals. Treatments included $CaCl_2$ at 0 (control), 50, 100, 150, or 200 µM, respectively. Control plants were

sprayed with distilled water. These sprays were applied in the morning hours (6 to 9 am). For each treatment, around 30 plants were used.

### 2.2. Determination of Photosynthetic Responses

#### 2.2.1. Determination of Calcium Treatment on Chlorophyll Fluorescence Parameters during Heat Stress

Chlorophyll fluorescence parameters include the minimal chlorophyll fluorescence intensity (Fo), maximal chlorophyll fluorescence intensity (Fm), variable chlorophyll fluorescence (Fv), maximum quantum yield of PSII photochemistry (FV/FM), nonphotochemical quenching coefficient (NPQ), and fluorescence decay ratio ($R_{Fd}$), which were measured according to [26]. The third leaflet was dark-adapted for 30 min and then observed by the quenching kinetics analysis method; the program settings for FluorCam (FluorCam 800MF, Photon System Inc., Brno, Czech Republic) were 50% actinic light (Act1), 20 s shutter speed, and 80% light intensity.

#### 2.2.2. Determination of Calcium Treatment on Chlorophyll Content during Heat Stress

Chl was measured according to [9]. Leaf samples (0.1 g) were homogenized with 1 mL of 80% ($v/v$) cold acetone, under dark conditions with maceration at 4 °C for 24 h. After centrifugation at 12,000× $g$ for 10 min, the reaction solution was read at 645 and 663 nm using a fluorescence plate reader (Infinite 200 PRO, Tecan, Grödlg, Austria). The amount of Chl was calculated according to the following formula:

$$\text{Chl content (mg g}^{-1}\text{ FW)} = (20.29 \times A645 + 8.05 \times A663) \times (Vt/W),$$

where Vt = final volume of extraction solution in mL, W = weight of sample in g.

#### 2.2.3. Determination of Calcium Treatment on Photosynthetic Indexes during Heat Stress

The photosynthetic indexes, including net photosynthetic rate ($An$), transpiration rate (E), and stomatal conductance ($gs$), were measured at 10:00 am on fully expanded leaves which were located in the middle and upper part of the branches and were not shaded by other plants or leaves. The parameters of the portable photosynthesis system (LI-6400, Li-Cor, Lincoln, NE, USA) were set according to [26]. The $CO_2$ concentration in the LI-6400 leaf chamber (Ca) was set at 400 µmol $CO_2$ mol$^{-1}$ air, the leaf temperature at 25 °C, and the RH of the incoming air at 65–70%. The flow rate was set to 500 µmol s$^{-1}$ [27], and photosynthetic photon flux density (PPFD) was 1800 µmol m$^{-2}$ s$^{-1}$ [28]. The water use efficiency (WUE, µmol mmol$^{-1}$) = net photosynthetic rate/transpiration rate [29].

### 2.3. Determination of Enzymatic Activity and Malondialdehyde (MDA) Content

The antioxidant enzyme activities were measured according to [30]. For crude enzyme extraction, the samples (0.25 g) were homogenized with 9 mL phosphate buffer saline (PBS, 50 mM, pH 7.8) and centrifuged at 10,000× $g$ for 15 min at 4 °C. The supernatant was utilized to determine SOD, POD, and APX activity.

The reaction solution for SOD activity consisted of 0.05 mL of enzyme extract, PBS (50 mM, pH 7.8), methionine (Met, 130 mM), nitro blue tetrazolium (NBT, 750 µM), ethylenediaminetetraacetic acid disodium salt (EDTA-Na$_2$, 100 µM), and 20 µM riboflavin. The reaction was incubated at 4000 lux for 20 min before reading at 560 nm, with PBS (50 mM, pH 7.8) as the blank. The SOD activity was calculated as follows:

$$\text{SOD (U g}^{-1}\text{ FW)} = Vt \times (A_{CK} - A_E)/(0.5 \times A_{CK} \times W \times V_s),$$

where $A_{CK}$ = reading of the blank reference, $A_E$ = reading of the sample, $V_s$ = determination volume of the reaction mixture solution in mL.

The reaction mixture for POD activity consisted of 1 mL of enzyme extract, PBS (50 mM, pH 6.0), 0.2% ($v/v$) guaiacol, and 30% ($v/v$) $H_2O_2$. The absorbance was read at

470 nm every minute for 3 times, with PBS (50 mM, pH 6.0) as the blank. The POD activity was calculated as follows:

$$POD \ (U \ g^{-1} \ FW) = (A470 \times Vt)/(0.01 \times t \times W \times V_s),$$

where t = reaction time in min.

The reaction mixture for APX activity included 0.1 mL of enzyme extract, PBS (50 mM, pH 7.0), ascorbic acid (AsA, 15 mM), and $H_2O_2$ (0.3 mM). The absorbance was read at 290 nm, with PBS (50 mM, pH 7.0) as the blank. The APX activity was calculated as follows:

$$APX \ (U \ g^{-1} \ FW) = (A290 \times Vt)/(0.01 \times t \times W \times V_s).$$

The MDA activity was determined according to [31]. The leaf samples (0.25 g) were homogenized with 5 mL of 5% (*w/v*) trichloroacetic (TCA) and then centrifuged at 10,000× *g* at 4 °C for 15 min. The supernatant (2 mL) was mixed with 2 mL of 0.67% (*w/v*) thiobarbituric acid (TBA) and then placed in a boiling water bath for 30 min. After cooling, the absorbance was read at 450, 532, and 600 nm. The result was calculated as follows:

$$MDA \ (mmol \ g^{-1}) = 6.45 \times (A532 - A600) - 0.56 \times A450.$$

### 2.4. Determination of Relative Electrical Conductivity (REC) and Osmotic Substance Contents

The REC was determined according to [32]. The leaves of plants of comparable size (to try to ensure the integrity of the leaves, fewer stem nodes) were taken. The leaves were washed with tap water for 45 s and then rinsed with distilled water three times, the surface water was blotted out with filter paper, the leaves were cut into long strips of suitable length (avoiding the main veins), and the fresh samples were quickly weighed into ten portions of 0.1 g each and placed in graduated test tubes with 10 mL of deionized water. The leaf samples (ten pieces) were bathed in water for 30 min at 25 °C. The initial electrical conductivity ($C_0$) was then read using a conductivity meter (DDSJ-318, Leici, Shanghai, China). The samples were then bathed for 5 min at 100 °C. After cooling, the final electrical conductivity ($C_1$) was read. The REC was calculated as follows: relative electrical conductivity (%) = ($C_1 - C_0$)/$C_0$ × 100%.

The soluble sugar content was determined according to [33]. After drying at 105 °C for 1 h and 70 °C for 24 h, the leaf samples (0.05 g) were homogenized in 80% ethanol. The solution (4 mL) was incubated for 40 min at 80 °C and then was centrifuged at 12,000× *g* for 1 min. The residue was repeatedly extracted with 4 mL of 80% ethanol, with each extraction combined with the supernatant. The supernatant was destained with 0.01 g active carbon at 80 °C for 30 min. The reaction mixture (1 mL of supernatant), 1.2 mL of $H_2O$, 0.005 g anthrone, and 3.8 mL of concentrated sulfuric acid were mixed. The absorbance was read at 625 nm, and the soluble sugar content was calculated according to a glucose standard solution curve and described as mg $g^{-1}$ dry weight (DW).

Soluble protein was extracted with coomassie brilliant blue [33]. Leaf samples (0.50 g) were homogenized with 5 mL PBS (50 mM, pH 7.8) and then centrifuged at 10,000× *g* at 4 °C for 15 min. The supernatant (0.1 mL) was mixed with 0.9 mL $H_2O$ and 5 mL coomassie brilliant blue. The absorbance was read at 595 nm. The soluble protein concentration was calculated according to a bovine serum albumin (BSA) standard curve and described as mg $g^{-1}$ fresh weight (FW).

The proline content was determined according to [34]. Samples (0.25 g) were mixed with 3% sulfosalicylic acid (*w/v*) and then placed for 5 min in a boiling water bath. The reaction mixture included 2 mL of supernatant, acid–ninhydrin reagent and glacial acetic acid and was placed for 30 min in a boiling water bath. After cooling, 5 mL of toluene was added and the absorbance was read at 520 nm. The proline content was calculated according to a proline standard curve and expressed as μg proline per g (μg $g^{-1}$).

### 2.5. Determination of Scanning Electron Microscopy (SEM)

The petals of Control and T2 groups were placed on pre-cooled slides, cut into small cubes of less than $5 \times 5 \times 2$ mm, and quickly fixed in 2.5% glutaraldehyde solution at 4 °C. The slides were then rinsed three times with phosphate buffer saline (PBS, pH 7.8) for 30 min and dehydrated three times with 30, 50, 70, 80, 90, and 100% ethanol gradients for 20 min each time. The samples were dried in a critical-point dryer (HCP-2, Hitachi, Tokyo, Japan), sprayed with gold, and photographed with an SEM (S-3400N, Hitachi, Tokyo, Japan).

### 2.6. Statistical Analysis

Statistical Analysis System (version 9.3, SAS Institute Inc., Cary, NC, USA) was used for ANOVA calculations. Sigma Plot (Systat Software, Inc., Chicago, IL, USA) was used for graphing data. The values are described as means ± SD, with different uppercase letters indicating significant differences ($p < 0.05$) according to Duncan's multiple range test.

## 3. Results

### 3.1. Effects of Calcium Treatment on Photosynthetic Responses during Heat Stress

3.1.1. Effects of Calcium Treatment on Chlorophyll Fluorescence Parameters during Heat Stress

As shown in Figure 1, we observed the chlorophyll fluorescence parameters of 'Carolla' leaves after external application of $CaCl_2$ under high-temperature stress. The larger and deeper the red area of the leaf in Figure 1, the higher its fluorescence. External $CaCl_2$ treatment resulted in an obvious decrease in the Fo of 'Carolla' compared to untreated plants under heat stress. Compared with the control, Fv and Fv/Fm were increased in T2 (100 µM $CaCl_2$), while Fm was not significantly different. In addition, $R_{Fd}$ increased slightly in T4 (200 µM $CaCl_2$) while NPQ and Fm showed a decline. In general, exogenous calcium treatment alleviated the effect of high-temperature on chlorophyll fluorescence parameters in 'Carolla'.



**Figure 1.** Effects of calcium treatment during heat stress on chlorophyll fluorescence parameters of 'Carolla' treated with 100 µM $CaCl_2$ (T2), 200 µM $CaCl_2$ (T4), or $H_2O$ as the control (CK).

3.1.2. Effects of Calcium Treatment on Chlorophyll Content during Heat Stress

Chlorophyll (Chl) is an important photosynthetic pigment in plants, and its content is closely related to light energy absorption, which can, to a certain extent, reflect the strength of photosynthesis. External $CaCl_2$ treatment resulted in an obvious enhancement of Chl content in comparison with untreated plants under high-temperature stress (Figure 2). In comparison with the control, the Chl content showed a 240 (T1, 50 µM $CaCl_2$), 358 (T2, 100 µM $CaCl_2$), 280 (T3, 150 µM $CaCl_2$), 167% (T4, 200 µM $CaCl_2$) increase with $CaCl_2$ treatment, while the rising trend of chlorophyll in T3 (150 µM $CaCl_2$) and T4 (200 µM

CaCl$_2$) gradually decreased with the increase in Ca$^{2+}$ concentration. Calcium treatment increased the Chl content, which likely affects the photosynthesis of *Rosa hybrida* 'Carolla'.

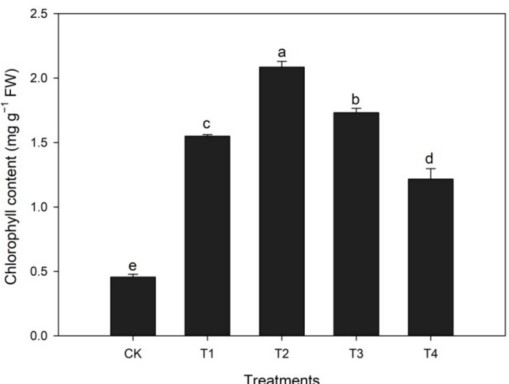

**Figure 2.** Chlorophyll (Chl) content in leaves of *Rosa hybrida* 'Carolla' under heat stress. Plants were treated with 50 μM CaCl$_2$ (T1), 100 μM CaCl$_2$ (T2), 150 μM CaCl$_2$ (T3), 200 μM CaCl$_2$ (T4), or H$_2$O as the control (CK). Each vertical line shows the standard error ($n = 10$). The values are described as means $\pm$ SD; different letters denote significant differences ($p < 0.05$) according to Duncan's multiple range tests.

### 3.1.3. Effects of Calcium Treatment on Photosynthetic Indexes during Heat Stress

External CaCl$_2$ treatment of 'Carolla' induced rising trends for the photosynthetic parameters of *An*, E, *gs*, and WUE in comparison with untreated plants under high-temperature stress. *An* was significantly increased by 2.26-fold (100 μM CaCl$_2$) compared with the control (Figure 3a). E showed a dose-dependent response and was increased by treatment with 50 μM CaCl$_2$ (77%), 100 μM CaCl$_2$ (141%), 150 μM CaCl$_2$ (122%), and 200 μM CaCl$_2$ (120%) (Figure 3b). Similarly, *gs* was significantly enhanced, by 0.50, 0.97, 0.75, and 0.63-fold, respectively, over the control (Figure 3c). WUE showed a 67% increase when treated with 100 μM CaCl$_2$ compared with the control (Figure 3d). Photosynthetic parameters increased as exogenous calcium concentration rose, peaking at an optimum concentration (T2, 100 μM CaCl$_2$) and then declining (Figure 3).

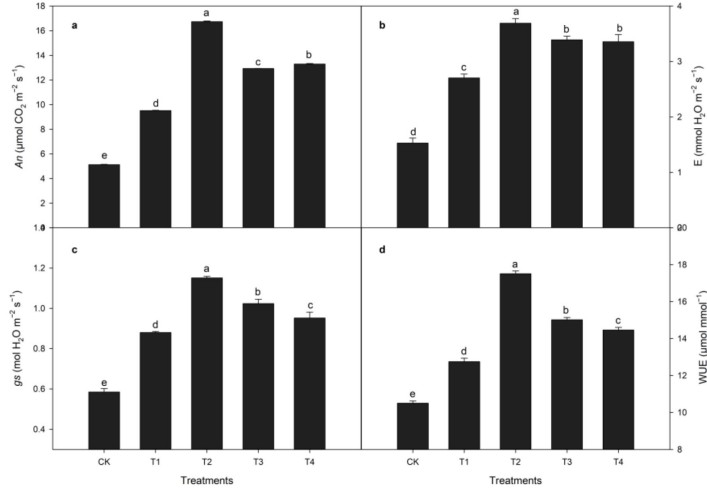

**Figure 3.** Effects of calcium treatment during heat stress on the (**a**) net photosynthetic rate (*An*); (**b**) transpiration rate (E); (**c**) stomatal conductance (*gs*); and (**d**) water use efficiency (WUE) in rose plants treated with 50 μM CaCl$_2$ (T1), 100 μM CaCl$_2$ (T2), 150 μM CaCl$_2$ (T3), 200 μM CaCl$_2$ (T4) or H$_2$O as the control (CK). Each vertical line shows the standard error ($n = 10$). The values are described as means $\pm$ SD; different letters denote significant differences ($p < 0.05$) according to Duncan's multiple range tests.

### 3.2. Effects of CaCl₂ Treatment on Antioxidant Enzyme Activities and MDA Activity during Heat Stress

In the groups treated with different concentrations of CaCl₂ solution, the activities of antioxidant enzymes (SOD, POD, and APX) significantly increased as compared to the control under heat stress. Specifically, the supplementation with 100 μM CaCl₂ (T2) resulted in a 52% increase in SOD activity (Figure 4b), a 68% increase in POD activity (Figure 4c), and a 47% increase in APX activity (Figure 4d) in comparison with untreated, heat-stressed plants. The content of MDA was significantly reduced by the CaCl₂ treatments, by 28% (T1, 50 μM CaCl₂), 62% (T2, 100 μM CaCl₂), 57% (T3, 150 μM CaCl₂), and 59% (T4, 200 μM CaCl₂) as compared to the control (Figure 4a). The decreased content of MDA indicate that lipid peroxidation in 'Carolla' caused by high-temperature was alleviated with calcium treatment.

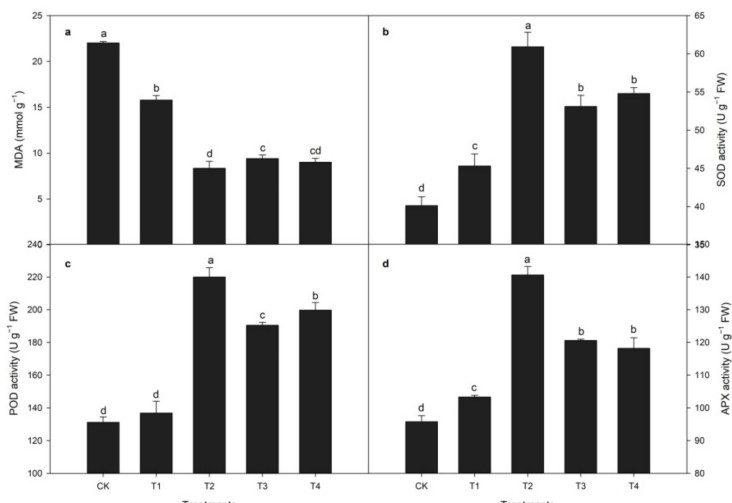

**Figure 4.** Effects of CaCl₂ treatment of different concentrations on the (**a**) MDA activity; (**b**) SOD activity; (**c**) POD activity; and (**d**) APX activity in *Rosa hybrida* under heat stress treated with 50 μM CaCl₂ (T1), 100 μM CaCl₂ (T2), 150 μM CaCl₂ (T3), 200 μM CaCl₂ (T4), or H₂O as the control (CK). Each vertical line shows the standard error (*n* = 10). The values are described as means ± SD; different letters denote significant differences (*p* < 0.05) according to Duncan's multiple range tests.

### 3.3. Relative Electrical Conductivity (REC) and Osmotic Substance Contents

External CaCl₂ treatment resulted in an obvious decrease in the relative electrical conductivity (REC) of 'Carolla' compared to untreated plants under heat stress. The CaCl₂ treatments significantly decreased the REC by 14 (T1, 50 μM CaCl₂), 30 (T2, 100 μM CaCl₂), 25 (T3, 150 μM CaCl₂), and 26% (T4, 200 μM CaCl₂) of the control (Figure 5a). The decrease in REC indicates that membrane systems in 'Carolla' remained better intact with calcium treatment under high-temperature. The osmotic substance contents increased with CaCl₂ supplementation under high temperatures. The proline content increased by 1.54-fold with 100 μM CaCl₂ (T2) in comparison with the control during heat stress (Figure 5b). Similarly, the calcium treatment increased the soluble protein contents by 0.16 (T1, 50 μM CaCl₂), 0.43 (T2, 100 μM CaCl₂), 0.25 (T3, 150 μM CaCl₂), and 0.25-fold (T4, 200 μM CaCl₂), respectively (Figure 5c). This tendency of increased osmotic substance contents was also reflected in the soluble sugar content (up to 225% with the 100 μM CaCl₂ treatment) in comparison with the control (Figure 5d).

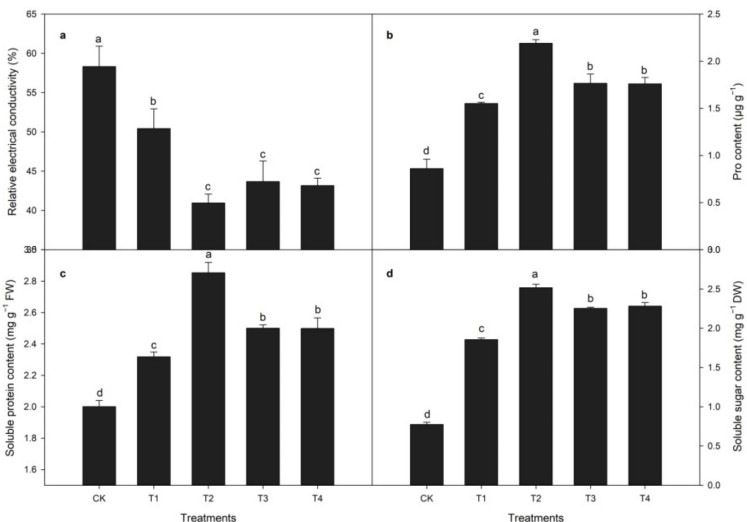

**Figure 5.** Effects of CaCl$_2$ treatment on the (**a**) relative electrical conductivity; (**b**) proline content; (**c**) soluble protein levels; and (**d**) soluble sugar content in *Rosa hybrida* under heat stress treated with 50 μM CaCl$_2$ (T1), 100 μM CaCl$_2$ (T2), 150 μM CaCl$_2$ (T3), 200 μM CaCl$_2$ (T4) or H$_2$O as the control (CK). Each vertical line shows the standard error (*n* = 10). The values are described as means $\pm$ SD; different letters denote significant differences (*p* < 0.05) according to Duncan's multiple range tests.

### 3.4. Effects of Calcium Treatment on Upper Epidermal Cells in Rosa hybrida 'Carolla' during Heat Stress

Under high-temperature stress, the conical–papillate shape of petal cells of 'Carolla' sprayed with water (control) was randomly deflated, loosely arranged, and the borders were blurred (Figure 6a,c). While exogenous calcium treatment, the conical–papillate shape of some cells was essentially in one plane, closely arranged, with clear borders of pentagonal or hexagonal cells (Figure 6b,d). Figure 6 shows that the defective cells in the field of view exposed to high-temperature stress were reduced after the addition of exogenous calcium.

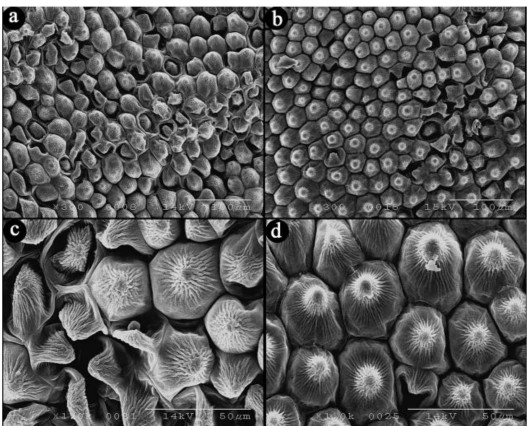

**Figure 6.** The anatomical structure of upper epidermis of 'Carolla' petals during heat stress from (**a**) control (CK, H$_2$O), 100 μm scale bar ; (**b**) T2 (100 μM CaCl$_2$), 100 μm scale bar; (**c**) control (CK, H$_2$O), 50 μm scale bar, and (**d**) T2 (100 μM CaCl$_2$), 50 μm scale bar.

## 4. Discussion

In plants that are periodically exposed to high temperatures, it is a common phenomenon for high temperatures to inhibit photosynthesis [35]. On the one hand (nonstomatal factors), the high-temperature reduces Rubisco activity which leads to PSII damage, and on the other hand (stomatal factors), the high-temperature causes stomatal closure, inducing the decrease in *gs*, E, and *An* [3]. Studies have shown that under high-temperature

stress, the pigment content and photosynthetic capacity of tomato [36] and the water use efficiency of Chinese cabbage [37] were significantly reduced. Under environmental stress conditions, exogenous calcium can strengthen the structure and function of the photosynthetic and respiratory organs in the mesophyll cells and reduce chlorophyll degradation [38], which alleviates the degree of damage caused by heat stress [24]. A previous study also reported that $Ca^{2+}$ acts as a secondary messenger for cytokinin action to promote Chl biosynthesis [39], which is consistent with the increase in Chl content in this study.

$CaCl_2$ protects chlorophyll and thus indirectly maintains the normal operation of the photosynthetic system. We observed that exogenous calcium treatment mitigated the adverse effects of high-temperature on chlorophyll fluorescence parameters including Fo, Fv, FV/FM, and NPQ, thereby maintaining PSII activity. This may be related to calcium treatment altering the redox status of primary quinone electron acceptor ($Q_A$) [40] or increasing photoprotective heat dissipation (NPQ) through the xanthophyll cycle [41]. High-temperature-stressed tobacco with $CaCl_2$ treatment diminished the concentration of $H_2O_2$, and therefore the plants could maintain higher $gs$ [42] as well as stomatal opening. Stomata adapt to environmental changes by regulating gas exchange, which in turn controls $CO_2$ absorption for photosynthesis and water loss for transpiration [43]. In addition, changes in $gs$ were positively correlated with changes in E and $An$ [40], thus contributing to the improvement of photosynthetic parameters. This may explain the upward trend in $An$, E, $gs$, and WUE of *Rosa hybrida* 'Carolla' by external $CaCl_2$ treatment reported in this paper. Exogenous application of $CaCl_2$ (10 µmol $L^{-1}$) increases the photosynthetic activities and membrane stability of tall fescue exposed to heat stress [44], $CaCl_2$ (10 mmol $L^{-1}$) increases the net photosynthetic rate ($An$) of *Agrostis stolonifera* [45], and $CaCl_2$ (20 mmol $L^{-1}$) alleviates the adverse effects of the net photosynthetic gas exchange of tobacco in high-temperature stress [42]. In *Rosa hybrida* 'Carolla', among treatments with different concentrations of $CaCl_2$, treatment with 100 µM $CaCl_2$ showed the best effects, with the photosynthesis of 'Carolla' reaching the maximum, and maximizing the recovery of chlorophyll fluorescence parameters, which can minimize the damage in 'Carolla' caused by high-temperature. $CaCl_2$ protected the chlorophyll pigment, indirectly maintaining the normal operation of the photosynthetic system.

Exogenous $CaCl_2$ treatment can protect the membranes of plant cells from oxidative damage [40]. $Ca^{2+}$ regulates the expression and synthesis of various antioxidant enzymes [46], such as catalase and glutathione reductase, thus increasing cellular levels of SOD [47], which acts as a front-line protector against superoxide radicals and further scavenged by APX via the ascorbate–glutathione pathway [48]. Studies have shown that the addition of exogenous calcium (12 mmol $L^{-1}$) increases the antioxidant enzyme activity of peanut [41] and wheat (10 mmol $L^{-1}$) [49] exposed to high-temperature stress. Under high-temperature stress, the SOD, POD, and APX activities of tobacco plants showed 18, 105, and 69% increases under treatment with 20 mmol $L^{-1}$ $CaCl_2$ compared to the control [42]. In *Rosa hybrida* 'Carolla', the SOD, POD, and APX activities increased by 52, 68, and 47% under treatment with 100 µM $CaCl_2$. Exogenous application of $CaCl_2$ (10 mmol $L^{-1}$) decreased the MDA by 29% in wheat leaves [50]. The reduction in MDA in plants treated with $Ca^{2+}$ is attributed to the binding of $Ca^{2+}$ to membrane phospholipids, which stabilizes the lipid bilayer and provides structural integrity [51]. In the present study, $CaCl_2$ treatment (100 µmol $L^{-1}$) decreased the MDA content by 62% and the REC by 30% in 'Carolla' exposed to heat stress. The addition of exogenous $CaCl_2$ increased the antioxidant enzyme activity, effectively countering the overproduction of reactive oxygen species (ROS) and the stress damage on cellular membranes caused by high temperatures.

Osmotic adjustment is one of the physiological indicators of the adaptive responses of species within global habitats [52]. Calcium, as a second messenger, not only protects the cell membrane structure [53] but also alleviates the adverse effects of numerous environmental stresses. Exogenous application of $CaCl_2$ (10 mmol $L^{-1}$) could regulate the contents of osmotic substances, subsequently alleviating waterlogging-induced damage to pepper plants [54]. Our results are in agreement with previous results [16], showing that

the proline, soluble sugar, and soluble protein contents were significantly increased while the relative electrical conductivity was significantly reduced with CaCl$_2$ treatment. The increase in SS and Pro is likely related to Ca$^{2+}$ enhancing the activity of starch metabolism enzymes such as α and β-amylases [40], as well as the activity of P-5-CR and γ-glutamyl kinase in the Pro biosynthetic pathway [55]. External calcium could alleviate cell membrane leakage [56] and osmotic stress, induce osmotic adjustment, and maintain plant growth under stress [57]. Research showed that Ca$^{2+}$ stabilizes the cell membrane surface, affects the pH of cells, and keeps solutes from leaking out of the cytoplasm [55]. In comparison to the groups treated with different concentrations of CaCl$_2$ solution and the control group, treatment with 100 μmol L$^{-1}$ CaCl$_2$ yielded the best results, with increases in proline content of 154%, soluble sugar content of 225%, and soluble protein content of 43%. The optimum concentration of calcium treatment for *Rosa hybrida* 'Carolla' is 100 μM CaCl$_2$.

Our study found that CaCl$_2$ sprayed on the leaves increased the heat tolerance of 'Carolla'. In addition, it has been shown that Ca$^{2+}$ and calmodulin (CAM) in leaves can be transported to flower buds and participate in flower bud differentiation of the plant, which in turn affects the development of petal cells and reduces the damage to petals caused by high temperatures [58]. High-temperature delays flowering and discoloration in ornamental plants [59], where flower color is affected by pigment, cell-sap pH, and metal elements [60]. It has also been reported that the petal epidermis cell shape can affect flower color [61]. Therefore, we observed the epidermal cells of petals in the control (CK, water) and optimum (T2, 100 μM CaCl$_2$) concentration CaCl$_2$ treatment groups. In this study, exogenous calcium treatment mitigated the disruption of petal epidermal cell morphology at high temperatures (Figure 6), while conical–papillate cells deepened petal color, improved saturation, and increased petal luster [61], thus reducing the effect of high-temperature on the flower color of 'Carolla'.

In summary, exogenous CaCl$_2$ increased SOD, POD, and APX activity of *Rosa hybrida* 'Carolla' exposed to heat stress, reducing MDA accumulation and relative electrical conductivity leakage to protect the integrity of cell membranes. At the same time, CaCl$_2$ maintained high photosynthetic capacity of 'Carolla' plants through strengthening the structure and function of the photosynthetic and respiratory organs and antioxidant enzyme activity. In addition, osmoregulators such as proline, soluble sugar, and soluble protein also play a better role in osmotic regulation by reducing osmotic stress and cell membrane leakage, protecting the integrity of the membrane and reducing high-temperature heat damage. Therefore, photosynthesis, antioxidant system activity, and osmotic regulation work together to regulate the formation for heat tolerance of *Rosa hybrida* 'Carolla'. Further, it has been shown that combined treatment with exogenous Ca$^{2+}$ and salicylic acid (SA) [4] or abscisic acid (ABA) [62] can provide better adaptation to heat stress. It has also been indicated that exogenous Ca$^{2+}$ enhances heat tolerance by initiating protein phosphorylation cascades to control transcription factors of specific stress-regulated genes, such as *HSFs* and *HSPs* [44]. A future goal is to build on the current foundation to clarify the signaling pathways involved in Ca$^{2+}$ to improve the heat tolerance of plants.

## 5. Conclusions

The results of this study prove that exogenous CaCl$_2$ decreased the damage to *Rosa hybrida* 'Carolla' by high-temperature. The physiological responses to high-temperature are reflected in photosynthesis, membrane stability, antioxidant enzyme activity, and osmotic regulator levels. Our study showed the positive effects of CaCl$_2$ in alleviating heat stress through upregulating photosynthesis, antioxidant enzyme activities, and adjusting osmotic regulation. These physiological responses are of great significance for improving the growth and yield performance of *Rosa hybrida* 'Carolla', especially in areas with high temperatures. The optimal concentration of exogenous calcium was 100 μM CaCl$_2$ for the alleviation of high-temperature stress, which provides evidence to support the use of exogenous calcium to improve the heat tolerance of *Rosa hybrida* 'Carolla'.

**Author Contributions:** Conceptualization, L.S., Z.W. and H.W.; methodology, H.W., Y.S. and K.W.; resources, W.S., S.H. and W.-S.K.; data curation, K.W., W.S. and S.H.; writing—original draft preparation, H.W. and Y.S.; writing—review and editing, L.S. and Z.W.; supervision, L.S.; funding acquisition, L.S. and Z.W. All authors have read and agreed to the published version of the manuscript.

**Funding:** This research was funded by the National Natural Science Foundation of China (No. 3217180536. "*RhPIF4*-mediated auxin signaling regulates bent peduncle phenomenon in rose (*Rosa hybrida*)").

**Institutional Review Board Statement:** Not applicable.

**Informed Consent Statement:** Not applicable.

**Data Availability Statement:** Not applicable.

**Conflicts of Interest:** The authors declare no conflict of interest.

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
