# Peer review of "Effect of Exogenous Calcium on the Heat Tolerance in Rosa hybrida ‘Carolla’"

_horticulturae, doi:10.3390/horticulturae8100980_

Round 1

Reviewer 1 Report

The paper Effect of Exogenous Calcium on the Heat Tolerance in Rosa hybrida “Carolla” by Han Wang et al. report that exogenous calcium improved the tolerance to high temperature in Rosa hybrida “Caolla” by enhancing photosynthesis and the antioxidant system, and

increasing osmotic substances. The results may be useful for horticulture field although the molecular mechanism how exogenous calcium affect photosynthesis, antioxidant system and the production of osmotic substances. There are several concerns and questions.

1. in Materials and methods section.

1) Plants were grown by hydroponic culture? If so, how much volume of culture media for each plant and how often was the media renewed?

2) How large are the plants (height and the number of leaves)?.

3) The authors mention the concentration of CaCl2 solution but not mention the total volume of the solution they gave to plant leaves (area of leaves?).

4) The nutrient solution contains Ca. It does not have any effect on the experiments?

2. Figure 1. shows the effect of calcium on the damage epidermal cells in petals. However, the main points of this paper are the effect of exogenous calcium on the photosynthesis, antioxidant systems, and accumulation of osmotic substances. These data were obtained from leaves. So, this petal data seems not related to other data.

1) author should mention why the epidermis of petal was affected by the applied Ca to leaves.

2)I think it is better to place the petal data in the last part because of above reason.

Minor

Figure 2. please explain the meaning of colors of leaves.

Reviewer 2 Report

Dear authors,

I have reviewed your manuscript "Effect of Exogenous Calcium on the Heat Tolerance in Rosa hybrida ‘Carolla’ submitted for publication in the journal Horticulturae. After reading the manuscript, I can tell that the you have put impressive work and effort into this. The manuscript is quite interesting and presents a valuable collection of information on how the exogenous calcium could improve the tolerance of Rosa hybrida ‘Carolla’ to high temperature stress.

I have listed a number of suggestions below:

Line 15: … by spraying four different.. In what part of the plant? Leaves?

Line 16: calcium chloride (CaCl2; 50, 100, 150, or200 μM)

Lines 17-18:… were measured under high temperature stress after….. When?

Line 23: The damages caused by high temperature stress were alleviated by ……What?

Line 28: Consider excluding “Optimal treatment concentrations”. You can add “photosynthesis

Line 31: Add the importance of the cut rose and especially for the hybrid Rosa 'Corolla'

Line 33: physiological, and biochemical changes

Line 35: The high temperature inhibits the photosynthesis [3],

Lines 38-39: temperature ranging from 15 °C to 28 °C is appropriate for the growth and flower production [6]. Heat stress….

Line 42: Add a reference

Line 47: Previous studies in 22 cultivars of commercial garden roses showed that…

Line 64: calcium (Ca2+)

Line 70: [22], and

Line 74: by spraying.. In what part of the plant? Leaves?

Line 85: 70-80%

Line 92: How many applications and for how long?

Line 93: at 0 (control), 50, 100, 150, or 200 µM.

Line 94: Control plants were

Line 97: The petals of CK and T2 groups. Please describe the treatments… The petal of control and ?? groups

Line 100: 30, 50, 70, 80, 90, and 100%

Line 112: method. The

Line 124: net photosynthetic rate (An), transpiration rate (E), and stomatal conductance (gs)

Line 125: fully expanded leaves. Was there a specific position on the plant?

Line 128: , and photosynthetic

Line 135: POD, and APX activity.

Line 137: ethylenediaminetetraacetic acid

Lines 161-162: The leaves were washed with tap water… How long?

Line 201: sprayed with water (control) was randomly deflated, loosely arranged, and the borders were

Line 209: Please replace CK for “control”

Line 214: As shown in Figure 2, we observed the

Line 217: Compared with the control,

Line 218: Describe “T2” and “T4”

Line 229: In comparison with the control, the

Line 230: Describe “T3” and “T4”

Line 241: you've described An, E and gs before

Line 244: with the control

Line 246-247: 0.75, and 0.63-fold

Line 248: with the control

Line 250: Describe “(T2)”

Line 261: d as compared to the control

Lines 265-266: Describe T1, T2, T3, and T4….. CK should be “control”

Lines 279-280: Describe T1, T2, T3, and T4….. CK should be “control”

Line 283 and 287: CK should be “control”

Line 285: Describe T1, T2, T3, and T4

Line 326: changes

Lines 341-342: [48], which

Line 345: 10 mmol

Line 346: 18, 105, and 69%

Line 348: 52, 68, and 47%

Line 363: soluble sugar, and soluble protein

Line 380: soluble sugar, and soluble protein

Line 384: Rosa hybrida ‘Carolla’.

References: Please only add the Journal abbreviations

Line 442: Camellia sinensis should be in italics

Reviewer 3 Report

Key words: “High temperature resistance” should be “high temperature tolerance”

Explain the tool for treatments' application in “2.1. Plant material and treatments”

Line 132. Please cite original references of the methods

Line 328 “Changes” should be “changes”

Mind the absence of gaps between the parenthesis and words in cited references throughout the manuscript text.
